# Cytoskeletal stiffening in synthetic hydrogels

Paula de Almeida[1], Maarten Jaspers[1], Sarah Vaessen[1], Oya Tagit [2], Giuseppe Portale [3], Alan E. Rowan[1,4] & Paul H.J. Kouwer [1]

Although common in biology, controlled stiffening of hydrogels in vitro is difficult to achieve; the required stimuli are commonly large and/or the stiffening amplitudes small. Here, we describe the hierarchical mechanics of ultra-responsive hybrid hydrogels composed of two synthetic networks, one semi-flexible and stress-responsive, the other flexible and thermo-responsive. Heating collapses the flexible network, which generates internal stress that causes the hybrid gel to stiffen up to 50 times its original modulus; an effect that is instantaneous and fully reversible. The average generated forces amount to ~1 pN per network fibre, which are similar to values found for stiffening resulting from myosin molecular motors in actin. The excellent control, reversible nature and large response gives access to many biological and bio-like applications, including tissue engineering with truly dynamic mechanics and life-like matter.

[1] Radboud University, Institute for Molecules and Materials, Heyendaalseweg 135, 6525 AJ Nijmegen, The Netherlands. [2] Radboud University Medical Centre, Radboud Institute for Molecular Life Sciences, Department of Tumor Immunology, Geert Grooteplein 26-28, 6500 HB Nijmegen, The Netherlands. [3] Zernike Institute for Advanced Materials, University of Groningen, Faculty of Science and Engineering, Nijenborgh 4, 9747 AG Groningen, The Netherlands. [4] The University of Queensland, Australian Institute for Bioengineering and Nanotechnology, Brisbane, QLD 4072, Australia. Correspondence and requests for materials should be addressed to P.H.J.K. (email: p.kouwer@science.ru.nl)

Hydrogels are attractive materials to mimic the complex microenvironment of cells and tissues[1–4]. The mechanical properties of the gels can be readily matched to those of target tissues or extracellular matrices[5], typically by changing polymer concentrations or cross-link densities. The vast majority of synthetic and biological hydrogels developed so far, largely display static mechanical properties, i.e., their physical properties do not change after hydrogel formation, although some will relax under stress[6–8]. Their static character is in stark contrast to nature where extra and intracellular matrix mechanics is far from constant. One of the simplest examples of dynamic stiffening in our daily lives is the actomyosin contraction. Myosin in an actin matrix forms the basis of muscular contraction, or (cortical) stiffening in nonmuscular cells, by transforming chemical energy stored as ATP into contractile stress[9]. Many other biological events also present stiffening over time, like tissue fibrosis[10,11] and tumour formation[12], or softening processes caused by enzymatic degradation[13]. Novel stiffening or softening approaches for (semi)synthetic networks commonly depend on in situ generation or removal of cross-links, but their application is still limited by the lack of reversibility[14–16], their unidirectionally (i.e., gels that only stiffen or soften[14,16]), by the long-time scales[16–18] and the small changes in modulus[16–19]. Despite these limitations, already significant biological stiffening effects have been observed.

Here, we report a biocompatible, yet fully synthetic, and dynamic hydrogel, composed of an interpenetrating network of the fibrous, stress-responsive polyisocyanide (PIC) and the thermoresponsive poly(N-isopropylacrylamide) (PNIPAM). Minute environmental changes (heating 1 °C) collapses the PNIPAM network, which generates stresses that transduce into a mechanical response and yields materials that become up to 50 times stiffer. The effect is instantaneous and fully reversible. The excellent control, reversible nature and large response gives access to many biological and bio-like applications, including tissue engineering with truly dynamic mechanical control.

## Results

**Interpenetrating double-responsive networks.** Ethylene glycol-substituted PIC gels uniquely mimic many aspects of biopolymer gels, but—as a synthetic polymer—can be tailored much easier[20–22]. The fibrous network architecture renders such gels soft, but strain-stiffening, i.e., their stiffness rapidly increases on deformation[23–25]. This effect, which is rarely found in synthetic polymers, is used here to generate large forces from small environmental changes. Force generation to deform the PIC network originates from the PNIPAM that undergoes a sharp morphological transition at its lower critical solution temperature[26] (LCST = 33 °C). As such, the energy source is temperature, rather than ATP in myosin motors. The hybrid or interpenetrating network (Fig. 1a) is formed by mixing the PIC polymer with the NIPAM monomer, cross-linker and initiator in cold water and heating the solution between 20 and 30 °C. Beyond its gelation temperature ($T_{gel} \approx 18$ °C), a dilute PIC solution (0.25–2 mg mL$^{-1}$) immediately forms a network of bundles, resulting in a soft gel. The thermally initiated polymerisation of NIPAM (30–500 mM) and the cross-linker $N,N'$-methylenebisacrylamide (MBAA, 0.5–3 mol% of NIPAM) is slower and, consequently, the PNIPAM network forms in between the PIC bundles[27]. This procedure yields an interpenetrating network (IPN) with both networks independently formed and only mechanically interlocked. Due to the presence of the semi-flexible PIC network, the resulting hybrid does not swell in water at $T > T_{gel}$ (Supplementary Fig. 1), and barely shrinks (<0.5%) when heated beyond the PNIPAM LCST (Supplementary Fig. 2). A constant volume is essential for 3D tissue engineering applications.

The transitions and mechanical behaviour are readily probed using rheology. Figure 1b shows the stiffness or storage modulus $G'$ of the PIC/PNIPAM hybrid hydrogel (concentrations 1.0 and 17 mg mL$^{-1}$, respectively), compared with both single networks at the same concentrations. At 17 mg mL$^{-1}$, the pure PNIPAM network (green data) does not percolate and is too soft to be

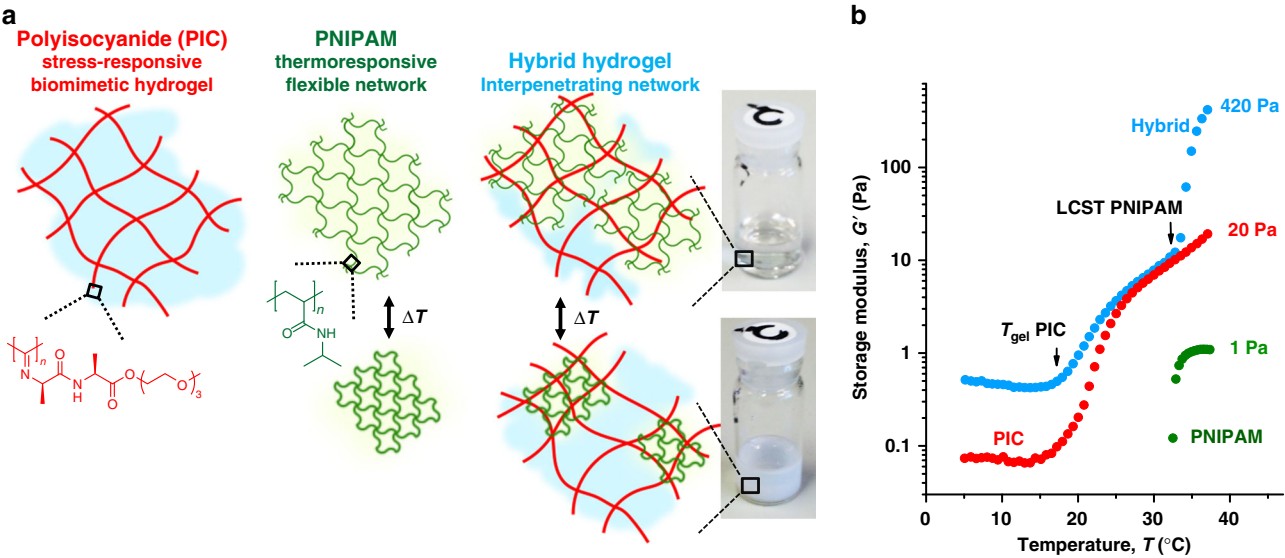

**Fig. 1** Controlling the stiffening response. **a** Schematic overview of the structure of PIC-PNIPAM hybrid hydrogels. The flexible PNIPAM network (green) is generated in the presence of the pre-formed semi-flexible PIC network (red), resulting in two interpenetrating networks. Heating the hybrid gel (transparent gel in the vial) beyond the LCST of PNIPAM ($T \approx 33$ °C) leads to a transition in the PNIPAM network. The hybrid becomes opaque, but does not shrink (no volumetric change). **b** Thermoresponsive mechanical properties of PIC-PNIPAM hybrid hydrogel with 1.0 mg mL$^{-1}$ PIC and 17 mg mL$^{-1}$ PNIPAM, and the single networks at the same concentrations. Arrows indicate PIC gelation and the PNIPAM LCST. The stiffness of the hybrid gel increases by more than an order of magnitude at the LCST of PNIPAM compared with the single component PIC hydrogel. The moduli of the PIC and hybrid gels are dominated by the elastic contributions. Loss data are provided in Supplementary Fig. 7

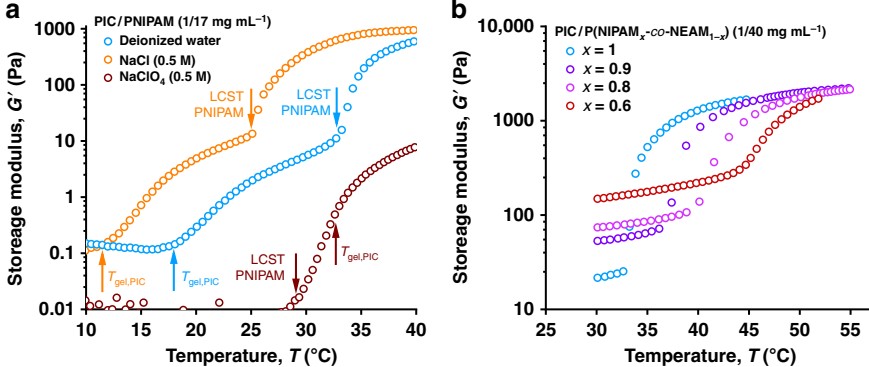

**Fig. 2** Controlling the stiffening temperature. **a** Shifting transition temperatures with salts; NaCl (orange data) shifts the PIC $T_{gel}$ and the PNIPAM LCST to lower temperatures. At 30 °C, the difference between deionised water (blue data) and 0.5 M NaCl is nearly a factor 100 in $G'$. NaClO$_4$ (red data) increases $T_{gel}$ and decreases the LCST, such that they reverse, which impedes the stiffening effect. The transition temperatures are indicated by the arrows. **b** Copolymerisation with NEAM shifts the LCST to higher $T$, whilst $T_{gel}$ remains constant (not shown). In addition, the transition broadens and the low-temperature modulus increases. At high $T$, the $G'$ reaches the same plateau for all copolymer compositions

reliably measured (Supplementary Fig. 3); only beyond its LCST, we observe a very soft material with $G' \approx G'' \approx 1$ Pa. The PIC gel (red data) forms at 18 °C, indicated by a jump in $G'$ and continues to stiffen with increasing temperature as a result of the increasing persistence length of the polymer chains and, hence, of the bundles[22]. Similarly, heating the hybrid PIC/PNIPAM gel (blue) shows PIC gel formation at 18 °C and a much larger jump in $G'$ at 33 °C. Within a few degrees, the hybrid is over 20 times stiffer than its single components.

The intense thermal stiffening response of the semi-interpenetrating gel stems from the strain-stiffening characteristics of the semi-flexible PIC gel. External deformation of this gel (and of other semi-flexible gels) is known to increase the modulus 10–100 times[28,29]. At the LCST, the collapse of the PNIPAM network in the hybrid similarly stresses the PIC network, but now the stresses are internal, rather than externally applied. We note that the collapse of PNIPAM in itself results in an increase in the gel stiffness[30], but that in our case, the nonlinear response of the PIC network amplifies this increase many times. The presence of internal stresses on the PIC network is supported by the development of a negative normal force $F_N$ (Supplementary Fig. 2), although the random orientation of the stresses and the scattered domains of the unpercolated PNIPAM fractions strongly reduce values for $F_N$ compared with earlier reported values for samples under uniaxial external deformation[31].

**Tailoring the stiffening temperature**. To realise the correct interpenetrating architecture, it is necessary to polymerise NIPAM above $T_{gel}$ of PIC (18 °C) and below the PNIPAM LCST (33 °C); properties of hybrids polymerised at 20 °C and 30 °C are nearly identical (Supplementary Fig. 4). Salt addition impacts both transition temperatures[32,33], following the Hofmeister effect, which already was used to predictably tailor the mechanical properties of PIC gels. A strongly kosmotropic salt like NaCl reduces both $T_{gel}$ and the LCST, which shifts the entire mechanical curve—with both transitions—to lower temperatures (Fig. 2a, orange data compared with blue data for deionized water). The extent is directly proportional to the salt concentration, for instance, both $T_{gel}$ and the LCST reduce ~3 °C in (kosmotropic) serum-free media (Supplementary Fig. 5). Chaotropic salts, such as NaClO$_4$ show different effects; they increase $T_{gel}$ of PIC, but decrease the LCST of PNIPAM. As a result, the polymerisation temperature window narrows, and at 0.5 M NaClO$_4$, $T_{gel} >$ LCST (brown data) and the PNIPAM network collapses before PIC bundles are formed, which inhibits transfer of the internal contractile forces and results in the loss of the

thermo-mechanical effect. To move the transition to physiological temperatures and beyond, we copolymerise NIPAM with N-ethylacrylamide (NEAM, 10–40 mol%), yielding copolymer LCSTs between 33 and 45 °C (Fig. 2b). With 10 mol% NEAM, the hybrid stiffens at exactly 37 °C. In addition, we observe that with increasing NEAM content, the low temperature modulus increases (the result of physical interactions between the PIC and the polyacrylamide networks), but the high-temperature modulus plateaus at the same level. The latter is fully in line with common strain-stiffening experiments that show high-stress moduli that are independent of concentration, temperature or polymer molecular weight[22].

**Network morphologies**. In the IPNs discussed so far, the PIC and PNIPAM-based networks are not covalently attached, but only mechanically interlocked. We disregard cross-linking between PIC and PNIPAM due to radical transfer reactions, since the PIC turns into a liquid at $T < T_{gel}$ (Fig. 3a, blue circles). One may argue that the stress transfer may be more effective when the two networks are covalently attached. Hence, we prepared a PIC gel where 3% of the monomers carries an acrylate functional group[2]. NIPAM polymerisation with this polymer introduces covalent bonds between the networks, which prevent disassembly of the PIC gel below $T_{gel}$ (Fig. 3a, green squares)[34]. Interestingly, the stiffening behaviour at the LCST is identical and the IPN is just as effective. Moreover, the IPN (with sufficiently low PNIPAM concentrations) offers the advantage of easy cell harvesting after liquefying the gel by cooling.

Similarly, changing the cross-link density (0.2–5 mol% MBAA) in the PNIPAM network does not affect the stiffening response considerably (Supplementary Fig. 6). In gels of PNIPAM only, the cross-link density controls the swelling capacity, but since the PIC do not swell in water (Supplementary Fig. 1), it is appreciable that the mechanical properties are independent of the cross-link density. In fact, even without cross-linker, i.e., only in the presence of linear PNIPAM chains, the PIC network stiffens at the LCST (Fig. 3a, orange diamonds). The thermo-mechanical transition at the PNIPAM LCST is much sharper for these semi-IPNs, which gives a 10-fold stiffness jump in no more than 0.5 °C. Additionally, these samples are readily prepared by simply mixing two cold polymer solutions at the desired concentrations, and, for potential biological applications, MBAA, a suspected harmful compound[35] is completely omitted. Note that for the semi-IPNs and the IPNs, a small loss modulus is found with $G'' \ll G'$ (Supplementary Fig. 7).

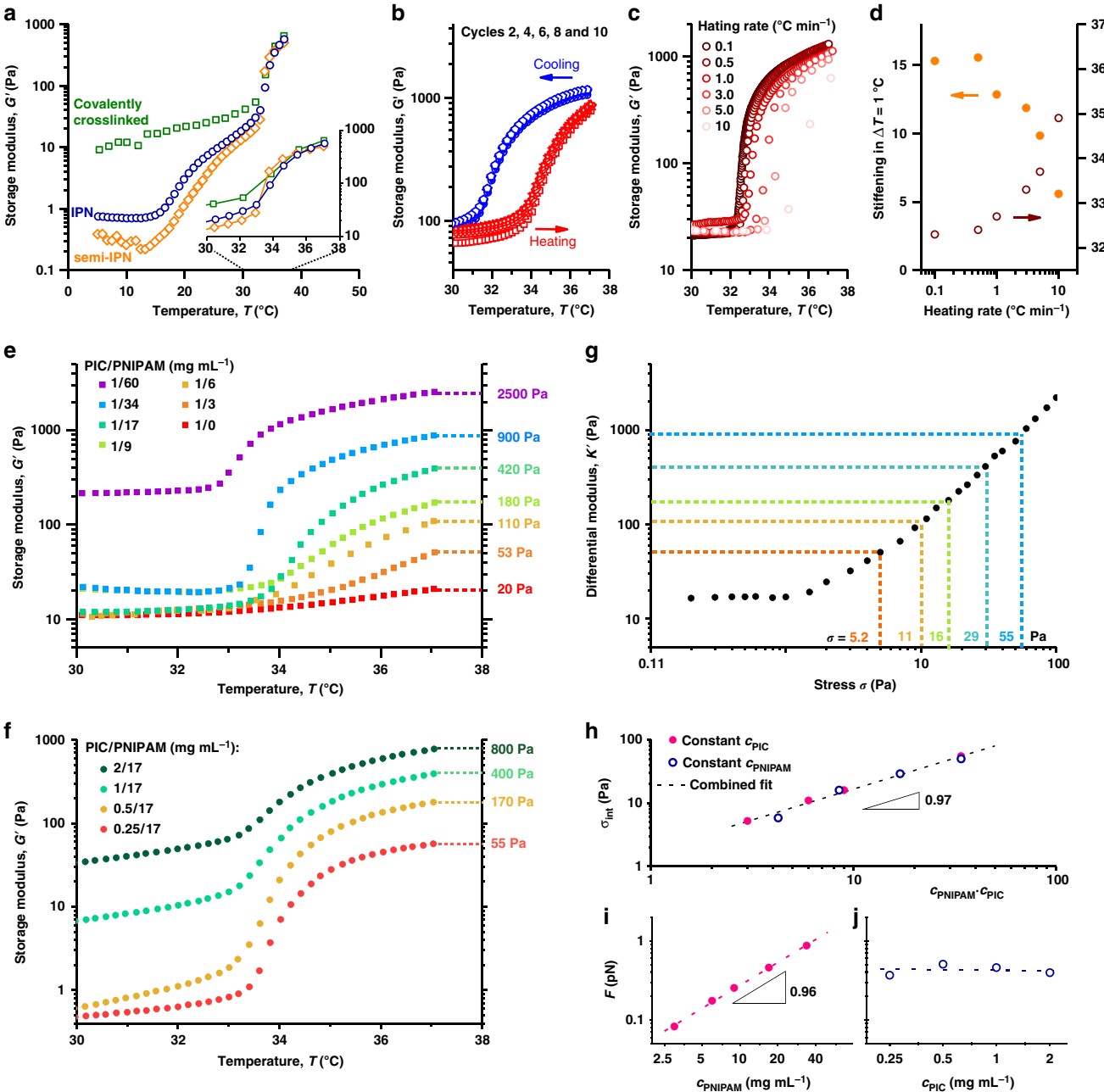

**Fig. 3** Stiffening of the PIC/PNIPAM hybrids. **a** Storage modulus $G'$ from different architecture hybrids. The stiffening response is identical for a covalently cross-linked network (green), IPN (blue) and a semi-IPN (orange). **b** Reversibility and small hysteresis observed after 10 cycles (only even cycles displayed). **c** Stiffening transition at different heating rates (0.1–10 °C min⁻¹). **d** Stiffening rates $G'_{T+1}/G'_T$ (orange) and transition temperature T (red) as a function of heating rate. The concentrations in **a**, **b** and **c**, **d** are PIC/PNIPAM 1/17, 2/17 and 1/40 mg mL⁻¹. **e**, **f** Increasing the PNIPAM (**e**) or PIC (**f**) concentration increases the storage modulus $G'$ of the PIC/PNIPAM hybrid after thermally induced stiffening. **g** Differential modulus K' against the external pre-stress σ for a 1 mg mL⁻¹ PIC hydrogel at $T = 33$ °C. The dotted lines correspond to the modulus of the PIC/PNIPAM hybrid gels at $T = 37$ °C and the external pre-stress σ corresponding to this modulus. Direct comparison between $G'$ and K' is allowed, because of the strong similarity between the experiments: in both cases, we determine δσ/δγ as a function of static pre-stress that is induced either by PNIPAM or by externally applied stress. **h** The average internal stress $\sigma_{int}$ generated by the PNIPAM network scales linearly with the product $c_{PNIPAM}·c_{PIC}$ for all studied samples. The dashed line is a power law fit to the experimental data (slope 0.97). **i**, **j** The average force per fibre as a function of $c_{PNIPAM}$ (**i**, $c_{PIC} = 1$ mg mL⁻¹) and $c_{PIC}$ (**j**, $c_{PNIPAM} = 17$ mg mL⁻¹) only depends (linearly) on the PNIPAM concentration (note the logarithmic x-axes). Lines are power law fits

**Response rates and reversibility**. The stiffening response of the hybrid is fully reversible; repeated thermal cycling (10 times) between 30 and 37 °C show excellent reproducibility and limited hysteresis (Fig. 3b). The response rate does depend on the experimental heating/cooling rate (Fig. 3c, Supplementary Fig. 8). Low heating rates (0.1 °C min⁻¹) showed very sharp transitions with a 15-fold stiffness increase in only 1 °C (for $c_{PIC}/c_{PNIPAM} = $ 1/17 mg mL⁻¹). Faster heating broadens the transition and shifts it to higher temperatures (Fig. 3d). Note that in our rheometer, the observed rates are limited by heat transfer from the Peltier plate to the aqueous sample. In slowly heated samples, one can readily evaluate the maximum degree of stiffening achievable over a pre-defined temperature window ΔT. For the intermediate concentrations $c_{PIC}/c_{PNIPAM} = 1/17$ mg mL⁻¹, we find that $G'$

increases 8-, 16- or 51-fold in a temperature window $\Delta T = 0.5$, 1 or 5 °C (Supplementary Fig. 9).

**Tuning the generated force**. The stiffening effect increases with the PNIPAM concentration, $c_{PNIPAM}$ (Fig. 3e). A denser PNIPAM network applies more internal stress to the PIC network and pushes the PIC stress-stiffening response to higher moduli; experimentally, we find $G' \propto c_{PNIPAM}^1$ ref. [4]. (Supplementary Fig. 10). Even at very low $c_{PNIPAM}$ (3 or 6 mg mL$^{-1}$), where the PNIPAM network certainly does not percolate but rather forms microgels, we still observe a clear, albeit small step in the stiffness of the hybrid (factors 2–5). When the PNIPAM stiffness dominates that of the stiffened PIC gel (at $c_{PNIPAM} > 60$ mg mL$^{-1}$), the amplification diminishes and we simply find the stiffening that one expects from PNIPAM gels[30] (Supplementary Fig. 10). Stiffening also increases with PIC concentration (Fig. 3f); in a denser PIC network, more bundles are impacted by the PNIPAM collapse. As PIC forms stress-stiffening hydrogels at very low concentrations[22], the mechanical response is still observed for $c_{PIC}$ as low as 0.25 mg mL$^{-1}$, with a 50-fold increase in stiffness at the PNIPAM LCST.

**Stress quantification**. We estimate the generated stresses by calibrating the samples to externally stressed PIC gels. At high stress, the mechanical properties of a gel are more accurately expressed by the differential modulus $K' = \delta\sigma/\delta\gamma$ (where $\sigma$ and $\gamma$ are the oscillatory stress and strain). At low stress, $K' = G'$, but beyond a critical value, the modulus increases rapidly: $K' \propto \sigma^m$ with the stiffening index $m \leq 3/2$ (Fig. 3g, black data)[22,28]. Comparing the moduli of the PNIPAM-stressed hybrid with the externally stressed PIC gel (Fig. 3g, dotted lines) allows us to

calibrate the average internally generated stress $\sigma_{int}$. Note that for the calibration, we restrict ourselves to $c_{PNIPAM} \leq 34$ mg mL$^{-1}$, since this regime, the modulus at 37 °C is dominated by PIC stiffening and the PNIPAM modulus is negligible. At higher PNIPAM concentrations, the latter starts to contribute significantly. Although theoretical work suggests that the mechanics of internal and external stress may be quite different[36], the analysis showed excellent agreement for internally and externally stressed networks of actin and myosin motors[9].

As we empirically find that the hierarchical stiffening effect requires the contribution of both materials in the network, we plotted the internally generated stress $\sigma_{int}$ against the product of the PIC and PNIPAM concentrations (Fig. 3h). All measured data points collapse to a linear master curve, which confirms that for the strongest effects, one needs both components sufficiently present. It is not trivial that we find such simple linear relation in a complex material that is intrinsically heterogeneous and nonlinear mechanically responsive. Apparently, however, the local differences in architecture are averaged over the entire sample to give a consistent macroscopic response. This averaging at larger length scales presents high reproducibility and great predictive power.

By averaging over the entire sample, we use the approximated $\sigma_{int}$ generated by the PNIPAM network to calculate the average force $F$ on a PIC bundle from $F = \sigma/\rho$, where $\rho$ is the (average) PIC bundle density in length per volume, analogous to the force analysis in actomyosin systems[20]. Obviously, for constant $c_{PIC}$, we retrieve a linear relation with $c_{PNIPAM}$ (Fig. 3i). More PNIPAM linearly increases the average force per PIC fibre. On the contrary, we find that the average force on a network fibre is independent of the PIC concentration (Fig. 3j). While the density of PIC fibres increases, the number of PIC fibres under stress increases as well

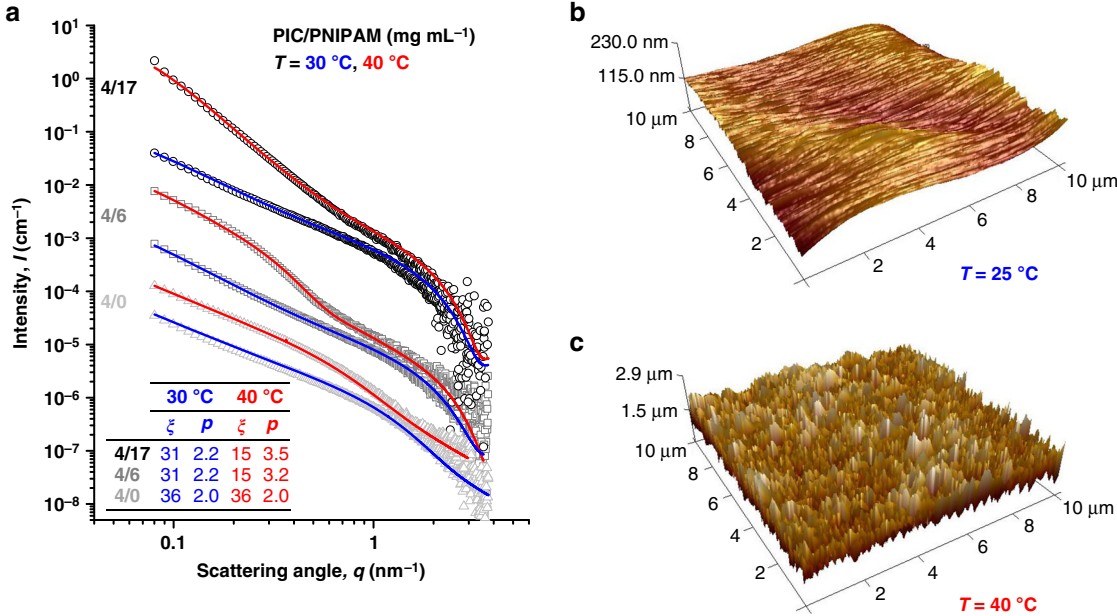

**Fig. 4** Microstructural characterisation. **a** Small-angle X-ray scattering profiles of PIC/PNIPAM hybrids (4/17 mg mL$^{-1}$ and 4/6 mg mL$^{-1}$) and single PIC networks at $T = 30$ (blue fit) and 40 °C (red fit) with PNIPAM contributions (Supplementary Fig. 13) subtracted. The high-angle concave curvature (in all samples) originates from the scattering contribution of the fraction of dissolved PIC chains not incorporated in the gel that is well described by a Kholodenko worm-like chain term with contour length $L = 72$ nm, persistence length $l_p = 8$ nm and radius $R = 1.0$ nm. The fits (solid lines) follow the correlation length model, with the exception of 4/6 at 40 °C, which also contains a Kholodenko bundle term with $L_B > 200$ nm, $l_{p,B} > 200$ nm (both outside the experimental window) and bundle radius $R_B = 6.7$ nm. The inset tabulates the key fitting parameters, correlation length $\xi$ (in nm) and Porod exponent $p$. Data are vertically offset for better visualisation (Supplementary Fig. 11, without offset). Note that in the SAXS experiments, the PIC network (4 mg mL$^{-1}$) scatters much stronger than the PNIPAM network (6 or 17 mg mL$^{-1}$), despite its lower concentration (Supplementary Fig. 13). To be sure that we only see the architectural changes in the PIC network, we subtract the (small) PNIPAM contribution. **b**, **c** AFM images of a PIC/PNIPAM (4 and 6 mg mL$^{-1}$) in water: at 25 °C with root mean square roughness $R_q = 20$ nm (**b**); and at 40 °C with $R_q = 325$ nm (**c**). Note the difference in z-scale for both images

and overall, the force per fibre remains constant. We anticipate that for high $c_{PIC}$, this independence no longer holds, but we are experimentally unable to reach this regime. Remarkably, the magnitude of the highest average forces found in the fibrous PIC network is quantitatively similar to the forces generated by myosin motors pulling on actin filaments in reconstituted cross-linked F-actin networks, where $F \approx 1$ pN was found[9].

**Mechanism of force transduction**. From small-angle X-ray scattering (SAXS) and atomic force microscopy (AFM) studies, we derived the mechanisms behind the stiffening response in the hybrids. SAXS scattering profiles of the PIC network below and above the PNIPAM collapse (Fig. 4a and Supplementary Fig. 11) show a major difference at small scattering angles $q$, due to the changed network architecture. For quantitative analysis, the profiles were fitted to the phenomenological correlation length model, used before[21] to describe the polydisperse PIC network: $I_{network}(q) \propto I_0[1 + (q\xi)^p]^{-1}$, where $I(q)$ is the experimental scattering intensity (Supplementary Fig. 12). The constant forward scattering intensity $I_0$ indicates that differences in scattering profiles should be attributed to changes in the network architecture (Supplementary Fig. 11). The fitting results show that as the PNIPAM collapses, the PIC network becomes denser (increase in the Porod exponent $p$) and the average pore size decreases (decrease in the characteristic network length scale $\xi$, see inset table Fig. 4), confirming that indeed, the PNIPAM collapse transfers to PIC network and activates the subsequent stiffening reaction. These architectural changes are fully in line with myosin-induced densification of F-actin networks[37] and may form the basis to precisely control mechanics in fully artificial life-like systems.

AFM imaging of the hybrid at $T = 25$ and $40\,°C$ (Fig. 4b, c) shows a large increase in the surface roughness of the gel, as the PNIPAM contraction forms cluster with dimensions of about ~100 nm. At low $c_{PNIPAM}$, they are isolated (highest roughness), but at higher concentrations, they merge and the roughness decreases (Supplementary Fig. 14). A cluster size larger than the PIC pore size is essential for an effective stiffening response, particularly at PNIPAM concentrations below the percolation threshold and, even more so, for linear PNIPAM chains. The absence of any stiffening response in recently published collagen/PNIPAM[38] hybrid may very well be attributed to this mismatch in length scales. To the best of our knowledge, no other hybrid network exhibiting similar hierarchical stiffening has been reported so far.

## Discussion

In summary, we prepared extremely responsive biomimetic hybrid networks composed of synthetic semi-flexible and responsive flexible polymers, which can become over 50 times stiffer in just a few degrees. Although the mechanism of force generation is completely different, the stiffening response is remarkably similar to what happens when myosin motor stress the F-actin network; even quantitatively similar forces are measured in the network. These stresses built up in the network linearly scale with the PIC and PNIPAM concentrations. The straightforward relation seems remarkable considering that the PIC network is highly heterogeneous, the PNIPAM even more when polymerised under the percolation threshold and considering that the PIC network shows a highly nonlinear mechanical response. Simulations on the mechanics of active materials[39] may soon be able to also provide further insight in multi-scale mechanisms of these complex materials.

The hybrids discussed here may be applied as dynamically switchable cytoskeletal components in synthetic cell-like systems[40] triggered by a small thermal cue, but the stimulus may be

replaced by any other mechanism that induces a collapse (coil-to-globule transition) of the flexible polymer component. More importantly, the 1 °C step allows one to control the mechanical properties in (PIC-based[24,41]) cell growth studies[42] where now, we are able to precisely and dynamically control the cellular mechanical environment[43]. In addition, at low PNIPAM concentrations, microgels and no percolating networks are formed, which favours 3D cell culture applications; in fact, cooling below $T_{gel}$ dissolves the entire hybrid into a low-viscous liquid that enables easy cell extraction. Towards advancing a true muscle mimic, we need to make the step from stiffening to macroscopic contraction, which requires unidirectional orientation of the semi-flexible polymer network.

## Methods

**Sample preparation**. Details of materials preparation are given in the Supplementary Information. Stock solutions were prepared in 18 MΩ cm purified water (MilliQ), unless mentioned otherwise. The hybrid hydrogels were prepared by mixing the PIC solution (4 mg mL$^{-1}$) on ice with a solution of NIPAM (2.0 M, Sigma Aldrich, recrystallized from a mixture of toluene and hexane (1:1) prior to use) and a solution of MBAA (0.1 M, Sigma Aldrich) in the desired ratios. For the samples involving salts, a stock solution (2.0 M) was diluted such that the final concentration amounted to 0.5 M. The molar ratio NIPAM:MBAA was fixed to 100:1, unless mentioned otherwise. To this mixture, potassium persulfate (PS, final concentration of 10 mM, Sigma Aldrich) and tetramethylethylenediamine (TEMED, 1 µL, Sigma Aldrich) were added to initiate the polymerization. The mixture was then transferred immediately to the rheometer and incubated for at least 1 h at $T = 30\,°C$ (default) or $T = 20\,°C$ (with NaCl) to allow the polymerisation reaction to complete. For the conjugated hybrid hydrogels, acrylate-functionalised PIC polymers (Supplementary Methods) were mixed with NIPAM, MBAA, PS and TEMED, and the final solution was immediately transferred to the rheometer, as described before. For the linear PNIPAM synthesis, an aqueous solution of NIPAM (350 mM) and PS (3.5 mM) at $T = 0\,°C$ was purged with N$_2$ for 15 min. Degassed TEMED was added and the polymerization was carried out at $T = 30\,°C$ for 1 h. At 0 °C, a mixture of PIC/PNIPAM was prepared to reach a final concentration of 1 and 17 mg mL$^{-1}$, respectively, and the solution was transferred to the rheometer for mechanical analysis.

**Rheology**. The mechanical properties were analysed on a stress-controlled rheometer (Discovery HR-1 or HR-2, TA Instruments) using an aluminium or steel parallel plate geometry with a plate diameter of 40 mm and a gap of 500 µm. The samples were loaded into the rheometer as a solution at $T = 5\,°C$ and immediately heated to $T = 30\,°C$. The complex modulus $G^*$ was measured in oscillation of amplitude $\gamma = 0.01$ at frequency $\omega = 1.0$ Hz. The samples were subsequently heated to $T = 37\,°C$ (1 °C min$^{-1}$). The DHR rheometers are equipped with axial force control and change the gap size when the axial force $|F_N| > 0.2$ N, which ensures good contact between the plates and the sample. The nonlinear response of the gel was evaluated by applying a constant pre-stress ($\sigma_0 = 0.2$–500 Pa) to the sample and a small superposed oscillatory stress ($\delta\sigma < 0.1$ $\sigma_0$; $\omega = 10$–0.1 Hz). From the measured oscillating strain $\delta\gamma$, we determine the differential modulus using $K' = \delta\sigma/\delta\gamma$. For the reversibility, a PIC-PNIPAM hybrid gel was analysed by multiple heating and cooling cycles between 30 and 37 °C with 5 min stabilisation between cycles. For the heating/cooling rate tests, the PIC-PNIPAM hybrid was cycled multiple times with rates 0.1–10 °C min$^{-1}$, also allowing for stabilisation between the cycles.

**Small-angle X-ray scattering (SAXS)**. SAXS measurements were performed at the BM26B station at the European Synchrotron Radiation Facility (ESRF), Grenoble, France. Samples were contained in 2 -mm quartz capillaries and inserted in a Linkam hot stage to control the sample temperature at $T = 30$ to 40 °C. The scattering intensity of only PNIPAM was low compared with that of PIC (Supplementary Fig. 13) and was subtracted from the double-network scattering, which yielded only the contribution of PIC network.

The obtained scattering intensities $I$ as a function of the modulus of the scattering wave vector $q$ were fitted to a combined Kholodenko worm-like chain model and the empirical correlation length model:

$$I(q) = I_{PIC}(q) + I_{network}(q) = (\Delta\rho)^2\varphi P_0(q, L, 2l_p)P_{CS}(q, R) + \frac{I_0}{1 + (q\xi)^p} \quad (1)$$

where $\Delta\rho = \rho_{polymer} - \rho_{water}$ is the electron density difference between the polymer chain (PIC) and the solution, $\varphi$ is the polymer volume fraction, $P_0$ is the form factor, $L$ is the bundle contour length, $l_p$ is the polymer persistence length (half of the Kuhn length), $P_{CS}$ is the cross-section form factor for flexible cylinders, $R$ is the cross-sectional radius of the polymer chain. In the correlation length model, $I_0$ is the forward-scattering intensity, $\xi$ is the characteristic length scale of the inhomogeneities and Porod exponent $p$ is associated to the network packing density.

In all PIC gels, a small fraction of the polymers is not incorporated in the gel and scatters as dissolved single semi-flexible polymers, which is described by a Kholodenko term with fixed fitting parameters contour length $L = 72$ nm, persistence length $l_p = 8$ nm and chain radius $R = 1.0$ nm, values obtained from earlier work[4]. The major component of the fitting model is the correlation length model that describes the network heterogeneities. PIC gels are comfortably fitted with $p = 2$, which is the limit where the correlation length model is identical to the Ornstein–Zernike model that we used earlier. For the PIC/PNIPAM (4/6 mg mL$^{-1}$) hybrid, not all PIC polymers are affected by the collapsing PNIPAM network at 40 °C. Therefore, we need a second Kholodenko term to describe this bundled fraction. For these bundles, we set $L$ and $l_p = 200$ nm (outside the experimental window) and find a radius $R = 6.7$ nm that is larger than the bundle sizes of the PIC network ($R_B = 2.9$ nm).

**Estimation of internal network forces**. From the estimated stress $\sigma_{int}$ generated by the PNIPAM network, we calculate the average force $F$ on the PIC fibres resulting from the thermal response of PNIPAM as:

$$F = \frac{\sigma_{int}}{\rho_{PIC}} \qquad (2)$$

where $\rho_{PIC}$ is the PIC bundle density in length per volume, which is defined as:

$$\rho_{PIC} = \frac{l_M N_A}{M} \frac{c}{N} \qquad (3)$$

in which $l_M$ is the length per monomer unit projected along the polymer backbone ($l_M = 0.25$ nm), $N_A$ is Avogadro's number, $M$ is the molecular weight of isocyanide monomer ($M = 0.316$ kg mol$^{-1}$), $c$ is the PIC concentration in the hybrid ($c = 1$ kg m$^{-3}$) and $N$ is the average number of polymers chains per polymer bundle ($N = 7.3$, determined by SAXS measurements[20]). Note that the analysis gives a macroscopic average, since both the PIC network and the PNIPAM structure are very heterogeneous.

**Atomic force microscopy (AFM)**. Atomic force microscopy images were obtained with a Catalyst BioScope (Bruker) coupled to a confocal microscope (TCS SP5II, Leica). Imaging was done in tapping mode using silicon nitride cantilevers with nominal spring constants of 0.06 N m$^{-1}$ (S-NL type D, Bruker). The samples were prepared on a WillCo-dish glass bottom dish and heated to 37 °C using a Lake-Shore Model 331 temperature controller. Water was kept on top of the gel to prevent drying out. Each sample was kept at this temperature for at least 1 h prior to measurements at $T > $ LCST. Root mean square (RMS) values were calculated using the roughness feature of NanoScope software (Bruker).

## Data availability

All data supporting the results of this study are available in the article and the Supplementary Information Files or from the corresponding author on reasonable request.

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

## Acknowledgements

The authors thank ESRF and the Dutch Science Foundation (NWO) for providing beam time at the Dutch-Belgian beamline (DUBBLE, grants BM26-02773 and BM26-02824). We thank the DUBBLE staff, in particular Daniel Hermida-Merino, for help during the SAXS experiments and Roeland Nolte for fruitful discussions. This project has received funding from the European Union's Horizon 2020 research and innovation Programme under the Marie Skłodowska-Curie grant agreement No 642687 (BIOGEL) and from the Dutch Ministry of Education, Culture and Science (Gravity program 024.001.035).

## Author contributions

P.d.A., M.J. and S.V. synthesized the polymers and did the mechanical characterization. P.d.A., M.J., S.V., A.E.R. and P.H.J.K. analysed the rheology data. O.T. conducted the AFM experiments. P.d.A., P.H.J.K. and G.P. analysed the SAXS data. P.d.A. and P.H.J.K. wrote the paper, with the input from all authors. P.H.J.K. supervised the project.

## Additional information

**Competing interests:** The authors declare no competing interests.

