## [Peer Review File · Nature Communications]

Reviewers' comments:

Reviewer #1 (Remarks to the Author):

In the revised manuscript, the authors have addressed some of the concerns I raised, largely through revisions of the text and moderation of the language and claims. This work does present an interesting new material with unique capabilities, and the advance of this work is in principle suitable for *Nature Communications*. However, there are a couple of remaining points that the authors need to address.

1. One point I am stuck on is the force balance of the hybrid hydrogel. If internal tensional stresses are being generated by contraction of the PNIPAM network, there should be corresponding external stresses to balance those internal stresses to prevent the gels from collapsing. I had asked whether the networks generated negative normal stress or why the hybrid gels were not shrinking/collapsing upon heating above the LCST (as was indicated in Fig. 1a of original manuscript). Is it just that the forces are so low overall, that they are below the measurement resolution of the rheometer or not sufficient to break the adhesion with the bottle? Perhaps the authors could do an estimation of what the force should be in their rheometer geometry? The authors should explain this. This is an important point because the forces are inferred, and not directly measured. Collapse/shrinkage of the gel, negative normal stress, or contractile stresses on adhesions to the bottle would seem to be a necessary outcome of the internal stresses indicated in Fig. 1a.

2. In my original comment 6, I had asked the authors to conduct a visualization of network architecture to further support the model drawn in Fig. 1a, using fluorescent labeling of either the PNIPAM or the PIC. The argument the authors make that only super resolution microscopy could visualize these network changes was not convincing, given the changes in structure indicated by the cartoon and the induced opacity observed as temperature is increased for the hybrid gel. With confocal fluorescence microscopy, the authors should be able to visualize homogeneity/heterogeneity of components at the scale of ~250 nm or so, and changes in homogeneity/heterogeneity as temperatures goes above the LCST. The authors should include confocal fluorescence microscope images of network architecture for the hybrid gels below and above the LCST with one of the components (or ideally both components) labelled.

Reviewer #1 comments on the authors' response to Reviewer #2:

The major points brought up Reviewer #2 were that the authors should create a muscle-like tissue with this material that actuates in order for the manuscript to be suitable for *Nature*, explore timescale of the stiffening, and describe volumetric collapse of the material as it stiffened. Further, reviewer #2 noted that the connection to actomyosin networks was superficial. The authors have not developed a muscle-like tissue, but my opinion is that this is not required for publication in *Nature Communications*. The authors have provided some data regarding the timescale of stiffening. While the authors have made some measurements of volumetric changes, the lack of volume collapse, as suggested by the cartoon in Fig. 1a and mentioned by the authors, is not observed for hybrid gels in the bottle or in the rheometer. This needs to be explained, and some description of the force balance in this system (i.e. how are internally generated tensional forces balanced out such that minimal external stresses are generated) should be provided, as I also ask for in my review. Finally, the authors have sufficiently scaled back the rhetoric about the similarity to actomyosin networks. Other minor points brought up by Reviewer #2 were satisfactorily addressed.

Reviewer #2 (Remarks to the Author):

In this revised manuscript, authors substantially refocused and clarified their message. The revised manuscript provides a type of stimuli-responsive synthetic gels that enable a quite large stiffening in quite narrow temperature interval. To do so, authors propose a robust system based on "hybrid" or "semi-IPN" formed by polyisocyanopeptide (PIC) and thermo-responsive chemically cross-linked poly(N-isopropylacrylamide) (PNIPAM) network. The underlying idea of this design is based on the analogy with biological systems (as actin network), but this version focus more on a materials science approach with the design of stimuli responsive synthetic polymer gels.

Most of the points raised by the reviewers have been clarified or discussed by the authors. After minor revisions, this manuscript should be published in Nature Communications.

I have several specific comments for the authors:

- In the abstract, the introduction (and the reply to comment #1), the authors state the novelty of their work by the strong thermo-responsive and reversible stiffening obtained without volumetric change (same polymer concentration). In the field, this issue has been already address by Guo, H., N. Sanson, D. Hourdet and A. Marcellan in "Thermoresponsive Toughening with Crack Bifurcation in Phase-Separated Hydrogels under Isochoric Conditions", *Advanced Materials* 28(28): 5857-5864 (2016). From my point of view, this reference could expand the scope of this work.

- In the reply to comment #2, authors emphasized on the fact that their "system is highly reproducible and predictable as expressed Fig. 3d." This should be state clearly in the manuscript. Fig. 3d requires further discussion.

The new section "Network Morphologies" has answered to comments #3, #4, #5 and #6. New data have provided in Suppl. Fig. S1, Suppl. Fig. 7 and semi-IPNs /IPNs discussion has been clarified.

The reference on Collagen/PNIPAm hybrid has been added.

Reviewer #1 (Remarks to the Author):

In the revised manuscript, the authors have addressed some of the concerns I raised, largely through revisions of the text and moderation of the language and claims. This work does present an interesting new material with unique capabilities, and the advance of this work is in principle suitable for Nature Communications.

Reply: We thank the reviewer for the evaluation.

However, there are a couple of remaining points that the authors need to address.

1. One point I am stuck on is the force balance of the hybrid hydrogel. If internal tensional stresses are being generated by contraction of the PNIPAM network, there should be corresponding external stresses to balance those internal stresses to prevent the gels from collapsing. I had asked whether the networks generated negative normal stress or why the hybrid gels were not shrinking/collapsing upon heating above the LCST (as was indicated in Fig. 1a of original manuscript). Is it just that the forces are so low overall, that they are below the measurement resolution of the rheometer or not sufficient to break the adhesion with the bottle? Perhaps the authors could do an estimation of what the force should be in their rheometer geometry? The authors should explain this. This is an important point because the forces are inferred, and not directly measured. Collapse/shrinkage of the gel, negative normal stress, or contractile stresses on adhesions to the bottle would seem to be a necessary outcome of the internal stresses indicated in Fig. 1a.

Reply: We feel that we did not explain sufficiently clear the changes in the hydrogel on PNIPAM contraction and subsequent stiffening. The contraction of the PNIPAM is not a macroscopically homogeneous contraction, like one would expect for a PNIPAM network only. Indeed such PNIPAM-only gel contracts into a lump of high-density material. In the IPN, however, macroscopic contraction is hindered by the PIC network: the PNIPAM contracts locally. The AFM experiments suggests that contraction takes place at the ~ 100 nm length scale, slightly larger than the mesh size of the PIC gel (although we do see the scattering response of the network, which indicates inhomogeneities at larger length scales). Adjacent to the contracted areas, the (water-filled) voids increases in size in the structure. For this very local mechanical deformation, no volumetric changes or significant normal stresses are expected, since, once more, no macroscopic deformation is applied to the sample.

As the reviewer suggested, we already measured the normal force in the hybrid at the PNIPAM transition. At low PNIPAM concentrations (17 mg mL^{-1} , i.e. below the percolation threshold), the normal forces are very small, indicating that indeed, the contraction really is local. At concentrations where the PNIPAM dominates the mechanical behaviour (60 mg mL^{-1}), we observe the expected macroscopic contractive behaviour (negative normal stress). Interestingly though, right at the PNIPAM transition, we first we see an initial increase in the normal stress.

To fully understand force balances in this complex and heterogeneous materials, we need some serious multi-length scales simulations, which is certainly not trivial, particularly not in semi-flexible networks [see: Lenz et al. in PNAS 2016 and Soft Matter 2015]. Simulations at this level is not something that we can do in our lab but in our outlook, we invite the specialist to contribute.

In the manuscript, we rephrased the section that discusses the mechanism and we modified the cartoon in Figure 1 to highlight the heterogeneity that is at the basis for volume conservation at the transition. We also refer to the normal stress measurements that we added to Supplementary Figure 2.

2. In my original comment 6, I had asked the authors to conduct a visualization of network architecture to further support the model drawn in Fig. 1a, using fluorescent labelling of either the PNIPAM or the PIC. The argument the authors make that only super resolution microscopy could visualize these network changes was not convincing, given the changes in structure indicated by the cartoon and the induced opacity observed as temperature is increased for the hybrid gel. With confocal fluorescence microscopy, the authors should be able to visualize homogeneity/heterogeneity of components at the scale of ~ 250 nm or so, and changes in homogeneity/heterogeneity as temperatures goes above the LCST. The authors should include confocal fluorescence microscope images of network architecture for the hybrid gels below and above the LCST with one of the components (or ideally both components) labelled.

Reply: Like for the normal force measurements, we also worked on this suggestion by the reviewer in the past months. We initially fluorescently labelled the PNIPAM, but did not obtain the desired resolution in CFM, also after heating beyond the LCST. Then, we labelled the PIC with fluorescent group, which is easier to image, due to its lower concentration. Indeed, STED clearly shows the PIC network (see Figure below: (a) at 30 °C, (b) at 40 °C). The denser structure supports our arguments, but we are hesitant to just put them in the manuscript, since quantitative reproducibility of the samples (even at different locations in the same sample) is difficult. This raises the general question: If we don't know exactly what we see, what then does it contribute besides a pretty picture? In this respect, we prefer not to use these results.

Reviewer #1 comments on the authors' response to Reviewer #2:

The major points brought up Reviewer #2 were that the authors should create a muscle-like tissue with this material that actuates in order for the manuscript to be suitable for Nature, explore timescale of the stiffening, and describe volumetric collapse of the material as it stiffened. Further, reviewer #2 noted that the connection to actomyosin networks was superficial. The authors have not developed a muscle-like tissue, but my opinion is that this is not required for publication in Nature Communications. The authors have provided some data regarding the timescale of stiffening.

Reply: We agree with the reviewer, no action needed.

While the authors have made some measurements of volumetric changes, the lack of volume collapse, as suggested by the cartoon in Fig. 1a and mentioned by the authors, is not observed for hybrid gels in the bottle or in the rheometer. This needs to be explained, and some description of the force balance in this system (i.e. how are internally generated tensional forces balanced out such that minimal external stresses are generated) should be provided, as I also ask for in my review.

Reply: see response to remark 1.

Finally, the authors have sufficiently scaled back the rhetoric about the similarity to actomyosin networks. Other minor points brought up by Reviewer #2 were satisfactorily addressed.

Reply: Many thanks, no action needed.

Reviewer #2 (Remarks to the Author):

In this revised manuscript, authors substantially refocused and clarified their message. The revised manuscript provides a type of stimuli-responsive synthetic gels that enable a quite large stiffening in quite narrow temperature interval. To do so, authors propose a robust system based on “hybrid” or “semi-IPN” formed by polyisocyanopeptide (PIC) and thermo-responsive chemically cross-linked poly(N-isopropylacrylamide) (PNIPAM) network. The underlying idea of this design is based on the analogy with biological systems (as actin network), but this version focus more on a materials science approach with the design of stimuli responsive synthetic polymer gels.

Most of the points raised by the reviewers have been clarified or discussed by the authors. After minor revisions, this manuscript should be published in Nature Communications.

Reply: We thank the reviewer for the evaluation.

I have several specific comments for the authors:

1. In the abstract, the introduction (and the reply to comment #1), the authors state the novelty of their work by the strong thermo-responsive and reversible stiffening obtained without volumetric change (same polymer concentration). In the field, this issue has been already address by Guo, H., N. Sanson, D. Hourdet and A. Marcellan in “Thermoresponsive Toughening with Crack Bifurcation in Phase-Separated Hydrogels under Isochoric Conditions”, *Advanced Materials* 28(28): 5857-5864 (2016). From my point of view, this reference could expand the scope of this work.

Reply: This is a good suggestion by the reviewer. In this beautiful example, the polymer design prevents shrinkage at the PNIPAM transition and stiffening of the collapsed network (due to reduced conformational freedom) can be readily measured. This is exactly what we observe at excess PNIPAM in the hybrid where mechanics is dominated by PNIPAM ($> 80 \text{ mg mL}^{-1}$). One should realise that in our material, the real stiffening is not the PNIPAM contraction, but the nonlinear (exponential) response of the PIC network to the PNIPAM contraction.

As suggested, we now refer to this work in the manuscript and we added the reference.

2. In the reply to comment #2, authors emphasized on the fact that their “system is highly reproducible and predictable as expressed Fig. 3d.” This should be state clearly in the manuscript. Fig. 3d requires further discussion.

Reply: We followed the reviewer’s suggestion: added the comment to the manuscript where we discuss Figure 3d (which is Fig. 3h-i-j in the revised manuscript).

3. The new section “Network Morphologies” has answered to comments #3, #4, #5 and #6. New data have provided in Suppl. Fig. S1, Suppl. Fig. 7 and semi-IPNs /IPNs discussion has been clarified. The reference on Collagen/PNIPAm hybrid has been added.

Reply: Many thanks, no action needed

REVIEWERS' COMMENTS:

Reviewer #1 (Remarks to the Author):

I appreciate the authors' responses to the two comments I raised in my last round of review. I support publication after consideration of the following minor comment. Regarding the imaging of the network architecture (comment 2): while the authors may themselves have already had an idea of what the network architecture looked like, I found the fluorescence images of the PIC network included in the response quite helpful. I would recommend that the authors include these images, as well as any images they have of the PNIPAM network (even if it just looks like a homogenous mesh) as a supplemental figure in the final manuscript. These data provide insight into the network architecture, a key aspect of this material, even if definitive conclusions about how the architecture changes across the LCST cannot be drawn. Scale bars should be included.

Reply to reviewer comments

Reviewer #1 (Remarks to the Author):

I appreciate the authors' responses to the two comments I raised in my last round of review. I support publication after consideration of the following minor comment. Regarding the imaging of the network architecture (comment 2): while the authors may themselves have already had an idea of what the network architecture looked like, I found the fluorescence images of the PIC network included in the response quite helpful. I would recommend that the authors include these images, as well as any images they have of the PNIPAM network (even if it just looks like a homogenous mesh) as a supplemental figure in the final manuscript. These data provide insight into the network architecture, a key aspect of this material, even if definitive conclusions about how the architecture changes across the LCST cannot be drawn. Scale bars should be included.

Reply: We appreciate the comments of the reviewer and share the desire to look at the mechanisms at the smallest possible length scales. Earlier, we already published the network structure of the PIC gel, based on AFM (dried in samples), SAXS and cryoSEM.¹⁻³ As said before, fluorescent microscopy images are in progress. We respectfully disagree that the initial results provided in the previous reply add much insight and we prefer to publish the data after we thoroughly understand what exactly we observe in these images. We decided not to add the microscopy images to the Supplementary Information.

References

1. Kouwer, P. H. J. *et al.* Responsive biomimetic networks from polyisocyanopeptide hydrogels. *Nature* **493**, 651-655 (2013).
2. Jaspers, M., Pape, A. C. H., Voets, I. K., Rowan, A. E., Portale, G. & Kouwer, P. H. J. Bundle Formation in Biomimetic Hydrogels. *Biomacromolecules* **17**, 2642-2649 (2016).
3. Schoenmakers, D. C., Rowan, A. E. & Kouwer, P. H. J. Crosslinking of fibrous hydrogels. *Nat. Commun.* **9**, 2172 (2018).